# Advanced Glycation End Products and Their Effect on Vascular Complications in Type 2 Diabetes Mellitus

**DOI:** 10.3390/nu14153086

**Published:** 2022-07-27

**Authors:** Jeongmin Lee, Jae-Seung Yun, Seung-Hyun Ko

**Affiliations:** 1Division of Endocrinology and Metabolism, Department of Internal Medicine, Eunpyeong St. Mary’s Hospital, College of Medicine, The Catholic University of Korea, Seoul 03391, Korea; 082mdk45@catholic.ac.kr; 2Division of Endocrinology and Metabolism, Department of Internal Medicine, St. Vincent’s Hospital, College of Medicine, The Catholic University of Korea, Suwon 16247, Korea; dryun@catholic.ac.kr

**Keywords:** diabetes mellitus, hyperglycemia, chronic complication, oxidative stress, glycation end products, advanced

## Abstract

Diabetes is well established as a chronic disease with a high health burden due to mortality or morbidity from the final outcomes of vascular complications. An increased duration of hyperglycemia is associated with abnormal metabolism. Advanced glycation end products (AGEs) are nonenzymatic glycated forms of free amino acids that lead to abnormal crosslinking of extra-cellular and intracellular proteins by disrupting the normal structure. Furthermore, the interaction of AGEs and their receptors induces several pathways by promoting oxidative stress and inflammation. In this review, we discuss the role of AGEs in diabetic vascular complications, especially type 2 DM, based on recent clinical studies.

## 1. Introduction

Type 2 diabetes mellitus (DM) has become increasingly prevalent over the past several decades, with an estimated prevalence of over 366 million by 2030 and over 693 million by 2045 [1,2,3]. Moreover, DM is a multifactorial and chronic metabolic disease caused by impaired metabolism of carbohydrates, fats, and proteins and ranks as the 11th leading cause of death caused by chronic complications worldwide [4]. DM is associated with morbidity and mortality due to its vascular complications. Recent studies have reported that young-onset type 2 DM (diagnosis at <40 years) has been correlated with an increased risk and higher burden of emerging complications [5,6]. As a result, the identification of diabetic complications is of considerable importance. Over the decades, research has been conducted on several alternative methods for preventing diabetic complications. Acute complications, such as hypoglycemia and hyperglycemia with ketoacidosis or hyperosmolar hyperglycemic status combined into the most serious acute life-threatening condition, can have a sudden onset [7]. In contrast, chronic complications are associated with the duration of DM (long-term exposure to hyperglycemia) and the degree of glycemic control and are categorized as microvascular complications, due to damage to small blood vessels, and macro-vascular complications due to damage to the arteries. Microvascular complications include diabetic kidney disease, diabetic retinopathy, and neuropathy. Diabetic complications such as coronary artery disease (CAD), cerebrovascular disease, and peripheral vascular disease (PVD) are categorized as macrovascular complications.

A near-normal level of intensive glycemic control helps to prevent diabetic vascular complications. Classic, established, and large-scale randomized controlled studies, such as the Diabetes Control and Complications Trial (DCCT) [8], United Kingdom Prospective Diabetes Study (UKPDS) [9], Action in Diabetes and Vascular Disease: Preterax and Diamicron Modified Release Controlled Evaluation (ADVANCE) [10], and Action to Control Cardiovascular Risk in Diabetes (ACCORD) [11], have investigated whether intensive glycemic control resulted in risk reduction for chronic microvascular complications. Intensive glycemic control, defined as glycated A1c (HbA1c) below 6.5–7.0%, contributes to a reduction in long-term outcomes of microalbuminuria, macroalbuminuria, polyneuropathy, and photocoagulation for DM retinopathy [8,9,10,11]. Early-stage control of DM reduces the long-term effect of hyperglycemic metabolic memory [9]. Thus, intensive glycemic control in the early stage of DM is emphasized as an effective preventive strategy against DM microvascular complications [8,9,10,11]. However, the association between intensive glycemic control and cardiocerebrovascular diseases is still not clear. In the majority of patients enrolled in these studies, the progression of complications from cardiovascular disease (CVD) was not prevented [12]. Therefore, for the prevention of CVD, consideration of multiple accompanying factors rather than only the glucose level has been emphasized. Recently, regarding DM management, multifactorial interventions, including blood pressure control, and maintenance of ideal body weight have been used [13]. Moreover, there is consensus for good glycemic control (HbA1c < 6.5%) and recommendations for individualizing HbA1c targets [14]. HbA1c, which is generated from the nonenzymatic glycation of hemoglobin, is the well-established standard for assessing glycemic management, and links between higher HbA1c and diabetic complications have been shown [15,16]. Because the mean HbA1c provides incomplete information about glycemic variability, long-term glycemic exposure, and fundamental causes of diabetic complications, markers of complications that account for the duration of hyperglycemia have been investigated [17]. Thus, the limitations of HbA1c lead to a focus on complementary methods for predicting diabetic complications [18].

In contrast to HbA1c, advanced glycation end products (AGEs) are products of glucose–protein or glucose–lipid interactions through glycation [19]. AGEs induce tissue damage. The mechanisms, pathogenesis, and consequences of AGEs on diabetic complications are diverse. The most crucial mechanisms involved in the progression of diabetic complications are induced by chronic hyperglycemia from decreased carbohydrate, protein, and lipid metabolism. Regarding hyperglycemia and its metabolism, chronic hyperglycemia induces an increase in oxidative stress, which is considered a factor of cellular, vascular, and tissue damage [20]. The concept of “metabolic memory” has been investigated from the perspective of nonenzymatic glycation of proteins, lipids, and nucleic acids [21]. The accumulation of AGEs in cells due to nonenzymatic condensation under oxidative stress conditions and activation of the polyol pathway, hexosamine pathways, and protein kinase C pathways is greater than that under physiological conditions, and AGE receptors are overexpressed. Via these pathways, various inflammatory cytokines are activated [22]. AGEs and their related molecules interact through crosstalk and alter the normal function of proteins [23]. Increased inflammatory cytokines and abnormal and stiffer glycated proteins contribute to the development of diabetic microvascular complications (retinopathy and nephropathy) and macrovascular complications. The concentration of circulatory AGEs was found to be increased in DM patients with CVD complications [24]. Thus, AGEs are the crucial molecules involved in the development of various diabetic complications.

Since AGEs were discovered in the early 1900s [25], the mechanisms underlying metabolic memory have remained unclear. There are various sources of AGEs, and few methods have been used to measure AGEs.

In this review, we summarize the process of AGEs formation and focus on the role of AGEs and AGEs receptors as pathophysiologic factors for vascular complications in type 2 DM based on clinical and experimental studies.

## 2. Sources and Formation of AGEs

AGEs originate from either endogenous or exogenous sources [26]. Approximately 30% of AGEs are absorbed into the systemic circulation via gastrointestinal absorption and the influence of the systemic burden. A high burden of AGEs is related to oxidative stress, inflammation, impaired innate defense, and insulin resistance [27]. Heterogeneous AGEs rely on the specific structure of protein-bound AGEs and are divided into protein-bound, peptide-bound, or free forms of AGEs. Unabsorbed AGEs in the colon are crucial factors for compromised glycemic control [28].

In contrast, AGEs are generated predominantly through endogenous processes via the nonenzymatic reaction of glucose-derived carbonyls with amino groups of lysine and arginine protein residues by forming unstable Schiff bases and stable Amadori products or fructosyl lysine, such as glycated hemoglobin, which is used in the diagnosis of DM, or follow-up modalities, and fructosamine [29]. This process, involving the formation of Schiff bases and Amadori products, is known as the Maillard reaction [25] (Supplemental Appendix A). Both of these compounds are reversible and react irreversibly with protein and peptide to form crosslinks. Then, levels of highly reactive α-dicarbonyl (DC) compounds, which are downstream products of Amadori and include glyoxal (GO), methylglyoxal (MGO), glycolaldehyde, and 3-deoxyglucosone (3-DG), are increased. These products are generated in DM but also under metabolic conditions other than nonenzymatic reactions [30]. α-DCs are inevitable in anaerobic glycolysis, the polyol pathway, and lipid peroxidation [31,32]. Because hyperglycemia in type 2 DM induces glucose toxicity due to higher glucose fluxes via the glycolytic pathway, excessive glycolysis gives rise to the accumulation of dihydroxyacetone phosphate due to insufficient triose phosphate isomerase activity and an increase in the formation of the highly reactive bicarbonyl MGO [33]. Lipid peroxidation, which is also increased in type 2 DM, results in lipid peroxidation end products (ALEs). The bicarbonyl products also have a role in the formation of AGEs [34].

AGEs have been categorized into two groups depending on their structure: the first group includes N-carboxymethyllysine (CML), pentosidine, crossline, pyrraline, and hydroimidazolone [35]; the second group includes AGE-1 (glucose-derived AGEs), AGE-2 (glyceraldehyde-derived AGEs), AGE-3 (glycolaldehyde-derived AGEs), AGE-4 (MGO/methylglyoxal-derived AGEs), AGE-5 (glyoxal-derived AGEs), AGE-6 (3-deoxyglucosone-derived AGEs), and acetaldehyde-derived AGEs (AA-AGEs) [36].

Thus, the formation and biochemistry of AGEs have been well-established elsewhere. AGEs formation extends the damage to macromolecules in tissue with structural and functional alterations [37]. Nevertheless, the mechanism of metabolic memory-related diabetic complications needs to be further investigated.

## 3. AGEs Interactions with Receptors

Exogenous AGEs and spontaneously produced endogenous AGEs interact through various signaling pathways and several AGEs receptors. AGEs bind into the extracellular transmembrane receptor and initiate signaling cascades. Among several receptors, the receptor for advanced glycation end products (RAGE) is a central transduction receptor of AGEs. RAGE, which is encoded on chromosome 6 near major histocompatibility complex III, is incorporated into a member of the immunoglobulin superfamily and recognized by its three-dimensional form rather than by specific amino acid sequences [38]. RAGE is expressed everywhere at a low level of RAGE ligand: endothelial cells, macrophages, monocytes, neurons, vascular smooth muscle cells, or chondrocytes [39,40]. However, RAGE is activated with an increase in the level of RAGE ligands in inflammation and its related responses. Thus, RAGE regulates inflammation via NF-ĸB, TNF-α, oxidative stress, and dysfunction of endothelial cells in type 2 DM [41]. In addition to the membrane-bound form of RAGE, there are two types of circulating soluble RAGEs (sRAGE) without transmembrane and cytoplasmic domains [42]. sRAGE is produced by cleavage of the cell surface receptor (cRAGE), which is generated by matrix metalloproteinases (MMPs) or by alternative splicing of endogenous secretory RAGE (esRAGE) [43]. Ligand binding enhances RAGE shedding, and serum sRAGE is considered representative of tissue RAGE expression [44]. Isoforms such as sRAGE and esRAGE bind RAGE ligands and block the interaction between membrane RAGE and cellular responses [45]. However, the exact function of sRAGE remains uncertain.

There is evidence of an increase in low-grade inflammation in type 2 DM. High levels of AGEs in type 2 DM patients are correlated with increased RAGE mRNA expression, protein carbonyl levels, and lipid peroxidation [46]. Moreover, the activation of signaling cascades, including NF-ĸB, and oxidative stress from AGE/RAGE interactions stimulate inflammation and tissue injury through the expression of vascular cell adhesion molecules, monocyte chemoattractant protein-1, endothelin-1, and plasminogen activator inhibitor-1 (PAI-1), and these factors are involved in vascular and tissue damage [47,48]. Therefore, AGEs with RAGE act as surrogate markers of inflammation, and their levels increase in chronic metabolic-inflammatory disorders [40].

In addition to RAGEs, the other classes of receptors are scavenger receptors, such as Stab1and Stab2, and AGE receptors (AGERs), such as AGE-R1~AGER [49,50]. These receptors can recognize and bind AGE ligands without the transduction of cellular signaling after engagement by AGEs. They have a role in the detoxification of AGEs and of AGE-specific ligand binding with degradation [51]. The expression of AGE-R1 was decreased, and AGE levels were elevated in DM patients [52,53]. AGE-R3 is hyperactive with hyperglycemia and high levels of AGEs [54].

## 4. Pathogenesis of Diabetic Vascular Complications and AGEs in Type 2 DM

Chronic complications of type 2 DM are caused by structural or functional modification of the vasculature. Structural modification results from extracellular or intracellular proteins or polypeptides that are vulnerable to modification by AGEs [55]. AGEs are found in the serum, the vasculature, the retina, and various renal compartments, such as the glomerulus and basement membrane [50]. Therefore, AGEs are involved in damage to multiple tissues or organs in type 2 DM after long-term exposure to hyperglycemia. Chronic hyperglycemia in uncontrolled type 2 DM accelerates the accumulation of AGE precursors, such as MGO, and activates the protein kinase C pathway, followed by increases in oxidative stress and inflammatory cytokine levels. Along with AGE accumulation, the AGE–RAGE axis is correlated with diabetic complications in patients with type 2 DM. Long-term DM complications are mainly categorized as microvascular complications, such as diabetic retinopathy (DR), nephropathy (DN), and peripheral neuropathy (DPN), and macrovascular complications, including cardiovascular disease, cerebrovascular disease, and PVD.

### 4.1. Microvascular Complications and AGEs

Microvascular complications are defined as the presence of retinopathy, nephropathy with albuminuria, and neuropathy [56,57]. Different organs are linked to these complications. DN is the leading cause of renal failure and is defined as estimated glomerular filtration rate (eGFR) < 60 mL/min/1.73 m^2^, and/or microalbuminuria > 3 mg/g, or an albumin-to-creatinine ratio (ACR) ≥ 3 mg/mmol in patients with DM [58,59]. The mechanism of DN is related to glomerular hypertrophy, renal oxidative stress, and fibrosis. Glomerular changes such as the thickening of tubular basement membranes, mesangial hypertrophy, and loss of podocytes are provoked by AGEs. RAGEs are also found in tubular epithelial cells and glomerular cells, including podocytes and mesangial cells. The activation of RAGE from AGEs enhances RAGE expression. Therefore, RAGE expression is prevalent where AGEs accumulate. Tubular cells are exposed to a large amount of AGEs and increase the activation of intracellular signaling pathways via their highly expressed RAGE [60]. This process is crucial for the development of interstitial fibrosis and glomerular dysfunction during the early phase in type 2 DM [60]. Therefore, recent studies have focused on the therapeutic effects of RAGE blockade in DN [61,62,63,64]. 

The AGE–RAGE pathway is activated via various signaling cascades, such as phosphoinositide 3-kinase (PI3K)/protein kinase B (PKB)/IκB kinase (IKK), and NF-κB activation [51]. NF-κB binds to the RAGE promoter and enhances RAGE expression. Increased levels of NF-κB in the kidney activate glomerular and tubular cell damage and induce renal injury. NF-κB also affects adhesion molecules and proinflammatory cytokines such as interleukin (IL)-6, tumor necrosis factor (TNF)-α, and monocyte chemoattractant protein (MCP)-1. These factors are involved in the development of DN. IL-6 is involved in pathologic changes in mesangial cells, and MCP-1 plays a role in mesangial cell proliferation. The AGE–RAGE interaction promotes reactive oxygen species (ROS), and ROS enhances the Janus kinase (JAK)-signal transducer and activator of the transcription (STAT) pathway [53,65,66]. JAK-STAT signaling has a crucial role in mesangial cells, podocytes, and epithelial cells and induces glomerular hypertrophy by inducing growth factors, including tumor growth factor (TGF)-β, platelet-derived growth factor (PDGF)-β, and IGF-binding protein-related protein-2 [67]. TGF-β regulates inflammation by upregulating MCP-1 and NF-κB [68].

AGEs themselves are involved in the progression of DN. AGEs form cross-links with matrix proteins such as collagen, leading to structural changes and inducing DM glomerulosclerosis with the accumulation of plasma proteins, lipid proteins, and immunoglobulin [69]. In addition to structural changes, the nonenzymatic glycation of type IV collagen provokes vascular permeability to albumin and the interaction with negatively charged proteoglycans [70]. In renal systems, the renin–angiotensin system (RAS), which comprises renin, angiotensinogen, angiotensin I, angiotensin-converting enzyme (ACE), angiotensin II, and their receptors, is a key factor in blood pressure and fluid balance. AGEs are involved in activated angiotensin II and trigger mesangial hypertrophy [70]. Angiotensin II also induces ROS production [71,72]. 

DR is characterized by abnormal vascular proliferation, which accompanies hemorrhage and ischemia in the retina. AGEs and CML are located in retinal vessels, and the levels are positively correlated with DR according to previous studies [54,73]. The AGE–RAGE pathway induces the apoptosis of pericytes and increases oxidative stress via NF-κB production. Increased levels of NF-κB upregulate vascular endothelial growth factor (VEGF) and allow endothelial permeability [74]. ROS generation, as in DN, can aggravate angiogenesis and vascular permeability. These pathologic changes cause damage to the subretinal membrane and microvasculature [75]. 

The role of AGEs in DPN is well established. DPN is stratified into the endothelium of the vasa nervorum, the sensory neuron (dorsal root), and Schwann cells [76]. As with DR, vascular dysfunction via the accumulation of AGEs in the endothelium of the vasa nervorum causes damage to the vascular structure and ischemia or occlusion. AGEs reduce the conduction of sensory and motor nerves and nerve blood flow [77]. The glycation of collagen and laminin induces alterations in the basement membrane and increases the permeability of vessels. Increased RAGE levels in dorsal root neurons activate the NF-kB cascade response [78]. The AGE–RAGE pathway promotes the intracellular activation of NADPH oxidase and the production of ROS. These pathological processes affect the majority of peripheral nerves [37].

#### Clinical Studies on Microvascular Complications and AGEs

Recent studies have investigated the association between AGEs and diabetic microvascular complications. Clinical studies have demonstrated that AGEs are positively correlated with the risk of microvascular complications (Table 1); however, the results in AGEs and complications remain inconsistent.

### 4.2. Macrovascular Complications and AGEs 

Atherosclerotic CVDs, including ischemic heart disease, cerebrovascular disease, and atherosclerosis, are leading causes of death worldwide, including in Korea [90,91]. Peripheral artery disease (PAD) with critical limb ischemia is also prevalent in type 2 DM. As previously mentioned, the etiology of DM complications is based on inflammation and changes in vasculature. Similar to the microvasculature, immune response and inflammation in vasculature are key factors in the pathogenesis of CVD. AGEs accumulation has been strongly related to cardiac pathophysiology. Recently, the role of AGEs in diabetic cardiomyopathy has been characterized by triggering the production of nitric oxide (NO) and inducing ventricular remodeling [92,93]. AGEs act in the progression of CVD through the modification of extracellular and intracellular proteins and signaling cascades via AGE–RAGE pathways.

#### 4.2.1. Role of AGEs in Macrovascular Complications

Among extracellular proteins, AGEs alter collagen, elastin, and laminin of the basement membrane and connective tissues [49]. Vascular stiffness is increased by the AGE crosslinking of collagen and elastin [24]. Glycated collagen changes endothelial cell activity and forms atherosclerotic plaques [94]. Laminin modified by AGEs alleviates binding to type IV collagen and inhibits the adhesion to endothelial cells for matrix glycoproteins. Thus, AGEs change the extracellular matrix function and integrity of arteries. AGEs can also alter lipids. Lipoproteins are targets of glycation. Low-density lipoproteins (LDLs) include forms of oxidized LDL, glycated LDL, and glycoxidized LDL. AGEs are linked to lipids by oxidative modification. LDLs are more often found in the form of glycated LDL in type 2 DM. Glycated LDLs can evade recognition from LDL receptors and can approach the arterial wall [95]. These glycated LDLs lead to intracellular accumulation and foam cell formation [96]. Glycated LDLs decrease the production of NO and clearance of LDL and promote atherosclerosis [97]. A previous study supported these mechanisms by investigating the elevated lipid-linked AGE levels in LDL both in vitro and in patients with DM [98]. Glycated high-density lipoproteins (HDLs) inhibit paraoxonase activity, which has a role in preventing LDL oxidation [49]. Glycated HDL also influences inflammation and reduces the removal of cholesterol and cholesterol transport, which are important processes in atherosclerosis [54,99].

#### 4.2.2. Role of AGEs in Cardiomyopathy and Atherosclerosis

Regarding intracellular AGE functions, AGEs induce crosslinking with intracellular proteins involved in Ca^2+^ homeostasis (impaired sarcoendoplasmic reticulum; SR Ca^2+^-ATPase pump) and result in cardiomyocyte dysfunction [100]. During the AGEs-RAGE interaction, Ca^2+^ levels are decreased by the upregulated ryanodine receptor, which has a role in balancing ion levels during the systolic and diastolic phases [101]. Overall, these pathologies result in the promotion of atherosclerosis.

The AGE–RAGE interaction upregulates mitogen-activated protein kinase (MAPK), PIP3K, p38, stress-activated protein kinase/c-Jun N-terminal kinase (SAPK/JK), and JAK/STAT signaling [53]. These cascades activate NF-κB and STAT3 transduction [102]. Thus, the overexpression of inflammatory factors promotes myocardial fibrosis. The AGE–RAGE interaction also influences vascular smooth muscle cells. Sakaguchi et al. demonstrated that smooth muscle cell proliferation upon arterial injury was suppressed in homozygous RAGE null mice compared with wild-type mice [103]. The crosstalk between oxidative stress and the AGE–RAGE axis is important in the context of diabetic macrovascular complications [104]. AGE–RAGE, as mentioned previously, results in signaling cascades downstream of MAPK, PIP3K, p38, and NF-κB and generates ROS with the acceleration of oxidative stress [105]. The AGE–RAGE axis in endothelial cells provokes the expression of genes such as p22phox and gp91phox, which are reduced forms of nicotinamide adenine dinucleotide phosphate (NADPH) oxidase and cause endothelial cell dysfunction. The prostacyclin in endothelial cells is inhibited by the AGE–RAGE system and promotes the de novo synthesis of PAI-1, thereby inhibiting fibrinolytic activity and then contributing to stabilization of the thrombus [106]. Therefore, therapeutic targets in this pathway have been studied, including blockade of RAGE, which reduces the development of atherosclerosis [107].

AGEs have a role in endothelial cell production of VEGF, which is involved in the development of atheroma. As aforementioned in DR and AGEs, the NF–ĸB–TNF–α–VEGF signaling cascade is activated by the AGE–RAGE system. The NF-ĸB pathway increases VEGR secretion, preventing the repair of endothelial lesions and inducing atherogenesis. Moreover, activated VEGF simulates differentiation from monocyte to macrophage in vasculatures and the accumulation of oxidized LDL, leading to foam cell formation [102]. Therefore, the AGE–RAGE system also activates pathologic inflammation in plaques and atheromas [105].

### Clinical Studies on Macrovascular Complications and AGEs

Clinical studies on the association between AGEs (and RAGE) and CVD in T2DM have recently been published (Table 2). Higher CML and AGEs levels showed an increased prevalence in PVD. Therefore, there has been evidence for a role of AGEs in diabetic complications. Koska et al. found an association between CVD risk and AGEs in a subgroup from the ACCORD trial. De la Cruz-Ares et al. assessed the difference in AGE levels between patients with CVD and established type 2 DM versus those with CVD and newly diagnosed type 2 DM [108]. Surprisingly, established type 2 DM patients showed higher levels of AGEs. However, previous studies could not prove a causal relationship. Stirban et al. demonstrated that nutritional AGEs showed direct detrimental effects on the vasculature [109]. In contrast to Stirban et al.’s result, Linkens et al. could not find a correlation between high AGE levels and vascular complications [110].

## 5. Summary

The pathophysiology of AGEs is closely related to glucose, lipid, and protein metabolism. Their metabolism is intricately entangled with oxidative stress and inflammatory reactivation according to chronic hyperglycemia. It is natural that AGEs are the crucial contributing factor to the progression of diabetic complications. Clinical and experimental research on the pathologic complications of DM has focused on AGEs as new biomarkers or therapeutic targets for several decades. Several clinical studies have determined that AGEs and RAGE are risk factors for vascular complications in type 2 DM. Nevertheless, the effects of AGEs and RAGE or other interactions on vascular complications among type 2 DM patients are not consistent in various studies due to the nature of the cross-sectional design or cohorts with a small number of samples. Therefore, there is a need for larger and longitudinally designed studies with validated detection tools for AGEs to make progress in the prevention of diabetic complications in the real world.

## Figures and Tables

**Table 1 nutrients-14-03086-t001:** Clinical studies—an association between microvascular complications and AGEs/RAGE in type 2 DM; Investigation of recent research within the past 10 years.

No	Authors	Year	Country	Subjects/Study Design	Finding
Diabetic nephropathy					
1	Skrha J. Jr., Soupal J., Loni Ekali et al. [79]	2013	Czechoslovakia	41 subjects with type 2 DM (47 subjects with type 1 DM) /cross-sectional	Higher AGEs levels were correlated with the albumin–creatinine ratio
2.	Galleau V., Cougnard-Gregoire A., Nov S et al. [80]	2015	France	418 subjects/cohort study	AGEs accumulation was associated with renal insufficiency
3	Yozgatli K., Lefrandt, J.D., Noordzij M.J et al. [81]	2018	UK	563 subjects/prospective cohort study	Development of microvascular complications was associated with HbA1c, not tissue accumulation of AGEs
4	Farhan S.S. and Hussain S.A. [82]	2019	Iraq	50 subjects Cross-sectional	There was a positive correlation between AGEs ratio and urine albumin/serum ratio in type 2 DM
5	Nishad R., Tahaseen V., Kavvuri R. et al. [83]	2021	India	130 subjects with albuminuria ranging from 150–450 mg/day/cross-sectional	A significant association between AGEs and impaired kidney function was observed in type 2 DM patients using AGE index.
6	Koska J, Gerstein H.C., Beisswenger P.J. et al. [84]	2022	USA	Action to Control Cardiovascular Risk in Diabetes (ACCORD) (n = 1150) and Veterans Affairs Diabetes Trial (VADT) (n = 447) participants/cohort study	AGEs in two different type 2 DM cohorts showed strong correlation with renal outcomes of reduced eGFR and macroalbuminuria.
7	Jin Q., Lau E.S., Luk A.O, Ozaki R, et al. [85]	2022	Hong Kong	3725 subjects/cohort study	AGEs measured by higher skin autofluorescence level were associated with kidney disease progression in type 2 DM
Diabetic retinopathy					
1.	Ng Z.X., Chua K.H., Iqbal T, et al. [86]	2013	Malaya	171 type 2 DM subjects versus 235 healthy control/case–control study	Proliferative DR patients had significantly higher levels of plasma pentosidine
Diabetic neuropathy					
1.	Vouillarmet J., Maucort-Boulch D., Michon P. et al. [87]	2013	France	66 subjects/prospective cohort study	AGEs measured by skin auto-fluorescence predict diabetic foot.
2.	Abuert C.E., Michel P.L., Gillery P. et al. [88]	2014	Switzerland	198 subjects/cohort study	CML and sRAGE were associated with DPN in patients with type 2 DM
3.	Zhao X.W., Yue W.X., Zhang S.W. et al. [89]	2022	China	560 subject/cohort study	Accumulation of AGEs measured using skin autofluorescence is correlated with DPN

AGEs, advanced glycation end products; CML, N-carboxymethyllysine; DM, diabetes mellitus; DPN, diabetic polyneuropathy; RAGE, receptor for advanced glycation end products.

**Table 2 nutrients-14-03086-t002:** Clinical studies—an association between macrovascular complications and AGEs/RAGE in type 2 DM; Investigation of recent research within 10 years.

No	Authors	Year	Country	Subjects/Study Design	Finding
1.	Stirban A., Kotsi P., Franke K. et al. [109]	2013	Germany	19 subjects/randomized-controlled study	Administration of a single AGE-modified protein impaired macrovascular function
2.	Chawla D., Bansal S, Banerjee B.D. et al. [46]	2014	India	75 subjects/cohort study	Serum AGEs levels were significantly higher in DM with vascular complications as compared to T2DM without complications
3.	De Vos L.C., Mulder D.J., Smit A.J. et al. [111]	2014	Netherlands	252 subjects/prospective cohort study	AGEs level measured by skin autofluorescence was significantly correlated with all-cause mortality and peripheral vascular disease
4.	Yozgatli K., Lefrandt J.D., Noordzij M.J. et al. [80]	2018	Netherlands	563 subjects in multicenter /cohort study	AGEs were associated with the development of macrovascular events
5.	Koska J., Saremi A, Howell S. et al. [20]	2018	USA	445 subjects from VADT and 271 subjects from the ACCORD study	Higher levels of select AGEs were associated with an increased incidence of CVD
6.	Ninomiya H., Katakami N., Sato I. et al. [112]	2018	Japan	115 type 2 DM and 25 type 2 DM subjects/prospective cohort study	AGEs can be utilized as a screening marker of atherosclerosis
7.	De la Cruz-Ares S., Cardelo M.P., Gutiérrez-Mariscal F.M. et al. [108]	2020	Spain	540 subjects/cross-sectional study	AGEs levels and intima-media thickness of the common carotid arteries were higher in patients with CVD and type 2 DM
8.	Linkens A.M., Houben A.J., Niessen P.M. et al. [110]	2022	Netherlands	82 subjects/randomized-controlled study	A 4-week diet low or high in AGEs had no effect on vascular function

AGEs, advanced glycation end products; CVD, cardiovascular disease; DM, diabetes mellitus.

## Data Availability

Not applicable.

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
