# Peer review of "Advanced Glycation End Products and Their Effect on Vascular Complications in Type 2 Diabetes Mellitus"

_nutrients, 2022, doi:10.3390/nu14153086_

Round 1
Reviewer 1 Report
1. Background of the study is not clear
2. I have found that that such type of similar paper is already available in public domain, authors needs to justify about the novelty in this current manuscript in comparison with existing papers.
https://www.ncbi.nlm.nih.gov/pmc/articles/PMC3951818/
https://www.ncbi.nlm.nih.gov/pmc/articles/PMC3903318/
https://academic.oup.com/jcem/article/93/4/1143/2826132
3. In the introduction authors needs to explain the role of protein glycation and formation of advanced glycation end products (AGEs) play an important role in the pathogenesis of diabetic complications like retinopathy, nephropathy, neuropathy, cardiomyopathy along with some other diseases.
4. In the introduction authors needs to explain in brief about intensive glycemic control affected the risk reduction for chronic microvascular complications.
Author Response
[July 22, 2022]
Editor-in-Chief: Maria Luz Fernandez and Lluis Serra-Majem
Nutrients
Dear Maria Luz Fernandez and Lluis Serra-Majem, Roy Zhang, and Editorial board members:
We appreciate your and the reviewers' efforts in reviewing our manuscript. In the revised manuscript, we have made corrections according to the reviewers' comments and suggestions. Responses to the referees’ suggestions have been listed one by one. Modified contents were written using the “Track Changes” function on MS Word. We included Figure 1 as the supplemental file according to reviewer 1’s suggestion. We sincerely expect that the enclosed revision will now be judged as acceptable for publication in Nutrients.
Thank you for your consideration. I look forward to hearing from you.
Sincerely,
Seung-Hyun Ko, MD, PhD
Division of Endocrinology and Metabolism, Department of Internal Medicine, St. Vincent's Hospital, College of Medicine, The Catholic University of Korea, Suwon 16247, Korea
Tel: +82-31-249-8900
Email: [email protected]

Reviewer 2 Report
This is a very interesting and useful paper, on the origin and actions of AGEs on the complications of diabetes mellitus.
The only suggestion that I can give concerns diabetic nephropathy. I suggest including, in section “4.1. Microvascular complications and AGEs”, a few words on the effects of AGEs on kidney tubular cells, since it was clearly shown that changes in these cells are very early events in diabetic nephropathy and precede glomerular functional and structural changes.
Author Response

(The authors gave the same response as above.)

Reviewer 3 Report
This review is well written, with all sufficient data and relevant literature. However, I have some things need to clarified.
1. In the abstract it is stated that the mechanism of how AGE is involved in diabetic vascular complications is investigated. However it is difficult to conclude about its mechanisms in the clinical research.
2. Please clarify how vascular endothelial growth factor (VEGF) in endothelial cells is involved in the development of atheroma?
3. It was suggested that AGEs with RAGE act as surrogate markers of inflammation, and their levels increase in inflammatory disorders. Please explain in which inflammatory disorders.
Minor comments:
Figure 1. can be put in supplemental files.
Row 155. there is space before the dot - inflammatory disorders [40] .
Author Response
[July 22, 2022]
Editor-in-Chief: Maria Luz Fernandez and Lluis Serra-Majem
Nutrients
Dear Reviewer:
We appreciate your and the reviewers' efforts in reviewing our manuscript. In the revised manuscript, we have made corrections according to the reviewers' comments and suggestions. Responses to the referees’ suggestions have been listed one by one. Modified contents were written using the “Track Changes” function on MS Word. We included Figure 1 as the supplemental file according to reviewer 1’s suggestion. We sincerely expect that the enclosed revision will now be judged as acceptable for publication in Nutrients.
Thank you for your consideration. I look forward to hearing from you.
Sincerely,
Seung-Hyun Ko, MD, PhD
Division of Endocrinology and Metabolism, Department of Internal Medicine, St. Vincent's Hospital, College of Medicine, The Catholic University of Korea, Suwon 16247, Korea
Tel: +82-31-249-8900
Email: [email protected]
